# Comparative Characteristics of Ductile Iron and Austempered Ductile Iron Modeled by Neural Network

**DOI:** 10.3390/ma12182864

**Published:** 2019-09-05

**Authors:** Borislav Savkovic, Pavel Kovac, Branislav Dudic, Michal Gregus, Dragan Rodic, Branko Strbac, Nedeljko Ducic

**Affiliations:** 1Faculty of Technical Sciences, University of Novi Sad, 21000 Novi Sad, Serbia (B.S.) (P.K.) (D.R.) (B.S.); 2Faculty of Management, Comenius University in Bratislava, 820 05 Bratislava, Slovakia; 3Faculty of Economics and Engineering Management, University Business Academy, 21000 Novi Sad, Serbia; 4Faculty of Technical Sciences Cacak, University of Kragujevac, 32102 Cacak, Serbia

**Keywords:** ductile iron, austempered ductile iron, cutting force components, artificial neural network, milling

## Abstract

Experimental research of cutting force components during dry face milling operations are presented in the paper. The study was provided when milling of ductile cast iron alloyed with copper and its austempered ductile iron after the proper austempering process. In the study, virtual instrumentation designed for cutting forces components monitoring was used. During the research, orthogonal cutting forces components versus time were monitored and relationship of cutting forces components versus speed, feed and depth of cut were determined by artificial neural network and response surface methodology. An analysis was made regarding the consistency of the measured cutting forces and the values obtained from the model supported by an artificial neural network for the investigated interval of the cutting regime. Based on the results, an analysis of the feasibility of the application of austempered ductile iron in the industrial sector with the aspect of machinability as well as the application of the models based on artificial intelligence, was given. At the end of the presentation, the influence of the aforementioned cutting regimes on cutting force components is presented as well.

## 1. Introduction

The development of modern machining certainly cannot be viewed without affecting the economic and ecological indicators of production. Reducing the use of cutting fluid is very important because the cost of cutting funds is 7–17% of total production costs. In accordance with that, implementation and development of sustainable production processes like dry machining or minimum quantity lubrication machining, is certainly interesting for research [1,2,3,4]. 

Application of dry or partially dry machining, requires measures to control the cutting fluid, switch the chip and regulate the adhesion between the tool and workpiece material [5]. 

In addition to the investigation of dry processing, further development directions have been taken to investigate difficult to cut materials and to measure their output characteristics during the chip removal process. Therefore, this paper is discussed on cutting forces and surface roughness [6], as well as tool wear [7,8]. Also, a wear test of the diamond-coated tool is in this study [9].

The direction of further research in the field of advanced production was based on the development of models of production process models. The development of these models was created in two directions: finite element modeling and artificial intelligence.

The application of finite element methods is presented in a study by Davoudinejad et al. [10] where the authors present a 3D simulation in micro-milling to find out the stresses in the tool and the shape of the chip. In addition, simulations of tool behavior are also presented in a study by Huo et al. [11] where the tool bending under the action of cutting forces is described analytically.

The direction of developing models based on artificial intelligence is more represented in production processes. A study by Sheikh-Ahmad et al. [12] used the mechanistic modeling process to foresee cutting force components and simulating the processing of polymers with fibers. Comparison of certain models of neural networks with experimental data showed good agreement. Application in modeling with use of artificial intelligence was presented in [13]. In order to maximize the economy of the face milling process, tool life tests are commonly tested when machining with a single tooth tool. It is possible to take advantage of these research results, if the radial or axial throw is within the permissible deviation interval (±0.002 mm). This prediction can be used for cutting forces [14]. The paper [15] presents a multi-criteria optimization approach, created by using a neural network, to determine optimal parameters during complex parts machining. A mathematical model is presented with spindle speed, feed rate, depth of cut and path spacing as the process output parameters are energy consumption and workpiece surface roughness parameters. Another study by D’Addona et al. successfully uses artificial neural network for prediction of the degree of tool-wear as a function of machining time [16]. The paper [17] compares characteristics of three methods for machine tools learning: polynomial regression, vector regression, and artificial neural network. Independent output cutting process parameters of cutting force, surface roughness, and tool life time were used during high-speed turning process. Alternative artificial neural network (ANN) models are certainly models based on genetic algorithms (GA). Thus, the paper by Mia et al. described the application of GA for modeling in the process of hard turning [18].

Austempered ductile iron (ADI) material is applicable in engineering construction. In addition, ADI material is also suitable for car construction, because of its strength and weight ratio as well as its wear resistance.

Good mechanical properties, such as wear resistance, tensile strength, and good ductility, make such material desirable in the automotive industry. Consequently, a better product is obtained, and on the other hand economic principles bring a positive balance in the production organization. A paper by Priarone et al. [19] discusses the results of milling experimental tests carried out on ADI 1050.

In other studies [20,21], a large number of experiments were performed, where the influence of austemperisation temperature as well as the length of the thermal process was investigated. ADI materials with different microstructures were used to investigate tool life.

As it is known, ADI material has good mechanical properties, but poor machinability compared to ductile iron and certain steels. The poor workability of ADI material is researched by simulating the tool wearing process [22].

In a study by Sadik et al. [23] it was determined that dry ADI material processing is better by 70–80% compared to machining with application of cutting fluid, with the application of a diamond coated tool.

The benefits of using ADI material are reflected in its good mechanical properties, such as good vibration properties, lighter weight compared to other structural materials, and uniform castability. The mechanical properties of ADI materials are closely related to the microstructural base, where the shape of graphite particles is the most prominent [24]. Two times more powerful than structural steel the ADI ductile iron classical quality, with other similar mechanical properties, which is primarily related to ductility and toughness. By adding titanium and boron material in the machining process with hard coated tools, it is possible to increase the hardness and abrasion resistance of the tools [25].

The study of the influence of alloying elements (Mo, Ni, and Cu) is given in a paper by Konca et al. [26]. It was found that Ni–Cu as an alloying element in the alloy GGG-60 with austemperation at 290 °C, contributed to increased hardness of the test sample by 44%.

These literary sources show that the influence of Ni and Cu as alloying elements is very large. This is first and foremost reflected in the temperature and time length of the austemperisation. [27].

The effect of influence of austempering temperature pearlitic ductile iron, with an emphasis on tribological characteristics was examined in [28]. Samples of ADI material were tested for adhesion and abrasion properties, using steel and ductile iron plates as the tribology couples during testing.

Additionally, monitoring cutting force can be used to extract machining information such as machine power requirements and moving parts loads. Measurement of cutting forces provides information about the machinability of materials and tool life. In general, chip cross section (feed and depth of cut) has been found to have the largest effect on cutting force or machining power. In the case of ADI, researchers have reported that the chip load is proportional to the cutting force.

Equation (1) calculates the cutting force based on the Kienzle–Victor relationship, which considers the chip cross section and unit cutting force *k*_*i*11_ depending on workpiece material and cutting force component; it is used very often:
(1)$$F_{i}=k_{i11}bh^{1-z}$$
where *b* and *h* are chip cross section parameters and *i* = *x*, *y*, *z* are cutting force components.

Equation (2), suggested by Taylor takes takes into consideration the influence of cutting speed:
(2)$$F_{i}=Cv^{x}s_{1}^{y}a^{z}$$
where the constant *C* depends on the workpiece material. This is in agreement with Equation (1).

Accordingly, this paper seeks to contribute to the study of cutting forces under different cutting conditions, by which both types of material can be successfully machined. In this range of research, which has not been shown in the literature presented, it is necessary to perform ANN-based modeling and adapt these models to the appropriate machine tools. The development of new materials certainly opens the space for creating databases with the most favorable cutting conditions in order to reduce tool consumption. This is controlled by appropriate values of the cutting forces and their monitoring in the material removal process. 

## 2. Experimental Research

The first objective of the experiment was to perform the measurement of cutting forces for different modes of ductile iron processing during face milling. In that order, output relationships for cutting forces and cutting conditions were determined according to the experimental plan of variation of the input parameters.

Material: In the study the following materials were used EN-GJS-700-2:
Nodular cast iron alloyed with 0.45% Cu (designated ductile iron (DI))ADI material, where DI austenized at 900 °C/2 h and austempered at 350 °C/2 h indicated (ADI)


The microstructure of nodular ductile cast iron (DI) EN-GJS-700-2 and austempered ductile iron (ADI) in a polished and etched state is given in Figure 1a,b. The same image shows the process of thermal processing of obtaining ADI material from DI, where the difference in the microstructure of the materials tested can be seen. The microstructure of ductile cast iron metal base consists of ferrite and mostly pearlite (with more than 90% pearlite) (Figure 1a). The microstructure of ADI material obtained after austempering of nodular ductile iron is ausferrite with a metal base, composed of acicular ferrite and austenite (Figure 1b). The amount of residual austenite in the ADI material microstructure is 16.56%.

Ductile cast iron and ADI mechanical properties are shown in Table 1. The austempering process improves all the mechanical properties of ductile cast iron, so it enhances the strength and hardness by 1.5 to 2 times. This increase in mechanical properties is caused by changing the microstructure of predominantly pearlite in the DI in ausferrite in ADI material.

### 2.1. Machine

The study was conducted on a vertical milling machine with 14 kW power. This machine was chosen because it was rigid enough and had sufficient strength. Tool: the milling head, Ø 125 mm, rake angle 0°, entrance angle 75°, was used in the study with the cutting insert of the HM P25 (hard metal of quality 25). The experiment was carried out with a single tooth and without cooling and lubrication agent. In this study the developed system for monitoring, acquisition, and measurement of cutting forces system with “Kistler” dynamometer type 9257 was used. This virtual instrumentation (VI) was developed by The Chair for Chip Removal Technologies, Department of Production Engineering, for cutting forces measurements.

### 2.2. Acquisition System for the Cutting Force Components during Face Milling Operation Monitoring

Figure 2 shows the scheme of the acquisition system for the cutting force components monitoring during face milling operation. Figure 2 shows that the acquisition system consists of the following components:
Machine tool (vertical milling machine)Tool (milling head with HM cutting inserts)Sensor of the monitoring system (three-component piezoelectric dynamometer, “Kistler”-9257A)Amplifier of monitoring system (capacitate amplifier “Kistler”-CA 500)Dial-up panel for connecting the module with the actual acquisition process (ED429-UP)Acquisition module A/D (analog-to-digital) converter-ED428Computer systemProgram (software) support systemVirtual instrument for acquisition, display in real-time, storing and processing data.


The virtual instrument used for measuring the three-component cutting force in face milling operation was developed by using the graphical software Lab VIEW 8.0. The virtual instrument (VI) is designed to allow for easy monitoring of voltage with dynamometer, which corresponds to the cutting force components F_x_, F_y_ and F_z_ values during the milling operation. VI displays time flow of cutting force components values in the form of diagrams and tables, and displays the maximal values of single tooth monitoring for one revolution of a milling cutter [14]. The front panel of the virtual instrument used in this study is shown in Figure 3; it is a maximal user-oriented concept designed using LabView software. The user can interrupt the measurement at any time if any of the three observed components falls outside the range of permissible values. At the time of exceeding the limits of cutting forces, tool wear usually occurs. The record with measured voltages and images is presented in the form of graphs and tables in the machining process (Figure 3). 

## 3. Neural Network Creation Architecture and Methodology 

When creating an artificial neural network architecture, several segments are considered. This primarily refers to input functions, which can be represented as continuous, binary or normalized data. This is followed by the transfer function and data collection as well as the learning methods. In addition, a method for identifying data as well as a mechanism for eliminating errors should be defined [29]. The relationship of the data used in the formation of the neural network is shown in Figure 4. The total amount of data used in the creation of networks was as follows: 80% represented the training (17 measurements), 10% was used for validation (2 measurements), and 10% was used for testing data (2 measurements). The figure below shows the obtained mean square error (MSE) as well as the regression coefficient (R).

In a two-layer neural network, the one of the two layers which has sigmoid data transmission is hidden, and the other is output (linear) and can arbitrarily select multidimensional data, that is, the experimental records on which network was created. Levenberg–Marquardt backpropagation is a network-based training algorithm (trainlm). In the event that memory expansion is required, a scaled conjugate gradient backpropagation (trainscg) is used. The selected network is Levenberg–Marquardt. This type of network requires more memory but less time for processing. The training process stops when the generalization stops improving, which is manifested by an increase in the mean error for data validation. The designed neural network consists of three input parameters, one hidden layer with 10 neurons and three output parameters. Since the output layer is created with a linear function, it must have the same number of neurons as there are three outputs. In the process of training the network, according Figure 5, the weight coefficient (w) and the correction factor (b) are changing. Three input parameters (cutting regime) and three output parameters (cutting force components) are considered. An architectural representation of the neural network flow chart with input and output parameters, as well as the functions within it, is shown in Figure 5.

## 4. Results of the Experiment and Discussions

For monitoring and data processing of the resulting cutting force components, response surface methodology of the three factorial designs of the experiment was adapted. This methodology in addition to making savings in the tool, workpiece material and time for trials, provides a sufficient reliable model between input and output parameters of the cutting process. That this methodology is good enough can also be seen in literary studies that dealt with the creation of appropriate regression models in the processes of material removal [13,14]. In addition to these simpler models, it is also possible to use extended models where the mutual influence of the parameters tested is highlighted [17,30]. 

The experimental results are presented using graphs. Figure 6, Figure 7, Figure 8 and Figure 9 show the cutting force components F_x_, F_y_ and F_z_ for the chosen cutting regimes for both materials. Due to different cutting conditions, the cutting force components change in the test interval, Equation (1). The graphs show that the components of the cutting force change according to the chip cross section and the rotation angle-cutting time. Figure 6 and Figure 8 show graphical records of the components of the cutting forces. Due to the narrow workpiece width, there is a large gap in the record of cutting forces. Therefore, for one revolution (e.g., 0.82 s to 1.1 s, Figure 6) the force record is only in the short cutting time range (0.82 to 0.84 s). The length of this record force increases with the width of the workpiece, which maximally moves up to the diameter of the tool, that is, in an angular record up to π radians (180°).

Figure 7a and Figure 9a show a combined record for three different cutting speeds in order to present the change in the components of cutting forces with increasing speed. It can be seen from the component F_x_ for both materials that it has the largest jump with increasing cutting speed, that is, it has the greatest influence on the resulting force F_A_. In the Figure 7b, a theoretical record of the components F_x_ and F_y_ is shown by half a revolution or 180°, as a representative with the maximum cutting width that can be achieved during this machining. It can be seen that the components for all different speeds in both materials largely follow the theoretic line of change of forces F_x_ and F_y_, respectively.

In Table 2 experimental cutting force component values for different cutting regimes and two investigated materials are shown.

Monitoring for 21 trials and for two types of workpiece material DI and ADI is presented in Table 2. Constants in models determined by response surface methodology for both materials and all three orthogonal cutting force components are shown in Table 3 and Table 4. The determined models by response surface methodology are adequate and all input factors are significant.

Cutting force component values for different cutting regimes and two researched materials (DI and ADI) determined by artificial neural network are shown in Table 5.

Figure 10 shows the regression coefficients of the training, validation, and testing of ANNs, as well as the cumulative regression. From this plot, the values of the regression coefficients are found to be more than 99.9%, which can be considered as very good results in model prediction. The regression coefficient of the described model examined is very well supported by studies [12,13] where the authors also calculated regression coefficients for the forces at milling process. In addition, we found similar regression coefficients in the measurements (i.e., modeling of other output characteristics of the material removal process), such as tool wear [8,16,17] and surface roughness [15,17].

Consequently, the regression coefficient has a high value, it can be concluded that the model is adequate.

Based on the analysis, it can be noted that, the values of the mean squared error and regression coefficient are very close to zero and one, respectively. Therefore, this concludes that the data match was very good. Figure 11 shows the histogram of error of the trained neural network in the process of training, validation, and testing. This figure also shows the errors in customizing the data, which are small, well-spaced, and close to zero.

Figure 12 compares the results of the following models: regression analysis, design of experiments, and neural network. Based on Figure 12, the lines for the creation of models and the criteria for the selection of models are based on Equations (3) and (4), which define the percentage deviation, as shown by histograms in Figure 13. The ANN created is applicable within the cutting force survey interval, that is, within the limits of the investigated cutting regimes (v = 44 ÷ 71 m/min; s_1_ = 0.178 ÷ 0.281 mm/t and a = 1 ÷ 1.83 mm). This threshold increases with the development of machine tools and cutting tools.

Figure 13 shows that the model based on the ANN has a better consistency of measured and model values, and therefore the error of deviation is smaller. It can also be seen that the deviation errors for ADI materials are smaller compared to DI, which is supported by the fact that the ADI material has more uniform microstructure compared to DI (Figure 1). In other words, when encountering nodes in DI machining, there are falls when measuring the cutting forces and, consequently, the unevenness of them, which is unfavorable for the tool life. This is another impetus for the application of ADI materials in manufacturing processes and in the industrial sector.
(3)$$E=\frac{\left|\mathrm{Model}\;\mathrm{by}\;\mathrm{regression}\;\mathrm{analysis}{}^{\prime \prime }\;\left(F_{x},F_{y},F_{z}\right)-\mathrm{The}\;\mathrm{measured}\;\mathrm{value}\;\left(F_{x},F_{y},F_{z}\right)\right|}{\mathrm{The}\;\mathrm{measured}\;\mathrm{value}\;(F_{x},F_{y},F_{z})}\cdot 100\%$$
(4)$$E=\frac{\left|\mathrm{Model}\;\mathrm{neural}\;\mathrm{network}\;(F_{x},F_{y},F_{z})-\mathrm{The}\;\mathrm{measured}\;\mathrm{value}\;(F_{x},F_{y},F_{z})\right|}{\mathrm{The}\;\mathrm{measured}\;\mathrm{value}\;(F_{x},F_{y},F_{z})}\cdot 100\%$$


Application of ANN finds the optimum value of the cutting regime in the interval of the tested cutting regimes, where three components are varied, and where it is possible to obtain the assumed cutting force for some regime that has not been experimentally investigated. It is also possible to supplement the neural network with some other input parameters if the need arises for this in the production process. An example is the introduction of a variable amount of cutting fluid. In experiments where there are multiple input factors, the application of artificial intelligence-based models certainly contributes. Finding the optimum is much more difficult as soon as the experimental design reaches beyond two factors. For illustration, dependence diagrams for the measured cutting forces are shown below.

Cutting forces component versus cutting speed, feed, and depth of cut for ductile iron are shown in Figure 14, Figure 15 and Figure 16. Cutting forces component versus cutting speed, feed, and depth of cut for ADI material are shown in Figure 17, Figure 18 and Figure 19.

In general, chip load or depth of cut and feed has been found to have the largest influence on cutting force components. In the case of ADI, researchers have reported that the chip load is proportional to cutting forces.

The effect of cutting speed on cutting force components can be seen in Figure 14 and Figure 17. The effect of the feed per tooth can be noticed in Figure 15 and Figure 18. It can be observed that when feed increases cutting force components increase as well. The effect of the depth of cut can be noticed in Figure 16 and Figure 19. As the depth of cut increases, all components of the cutting force increase significantly. Higher values of cutting force components are noticeable for ADI material.

Growth of the market for the application of ADI materials is on the rise, but the limiting factor is a heavy machinability. Based on the analysis of workability in the literature, three different strategies for processing parts from the ADI material are distinguished [28]. The first option is to perform rough treatment before austempering. Then the heat treatment is done after finishing the operation. This option is feasible when there is a wide tolerance of the final measures and the quality of the surface of the workpiece. Another option is to complete the machining before heat treatment. This option is feasible if it is possible to accurately assume the expansion of the material after an austemperization. The final, most profitable option is to machining parts after heat treatment. Choosing some of these strategies certainly affects the economy of the machining process.

## 5. Conclusions

Analysis of cutting force during milling operation is very complex due to the influence of a number of different phenomena. The study indicates the complexity of the processes that take place during cutting. The highest values of cutting force components have F_x_, medium F_y_ and at least F_z_ component. As the cutting speed increases, the influence of the force component of the cut F_x_ remains dominant over the other two components. Contributing to this knowledge is a monitoring system where irregularities in the cutting process can be detected at any time.

Strong correlations relating to cutting force components in relation to time are determined during the dry milling operation.

Higher values of cutting force components are more noticeable for ADI than DI material.

Artificial neural network models for both tested materials are determined in the tested interval. Based on the planar and spatial view, it is possible to determine the optimum machining regime for the set cutting conditions (v = 44 ÷ 71 m/min; s_1_ = 0.178 ÷ 0.281 mm/t and a = 1 ÷ 1.83 mm).

When feed and depth of cut are increasing, cutting force components significantly increase. Higher values of cutting force components are noticed for ADI material. Cutting speed is less affective on cutting force components values.

The model based on the ANN has a better consistency of measured and model values, and therefore the error of deviation is smaller for both materials in study. This is supported by the fact that the ADI material has more uniform microstructure than the DI where the nodules are separated and larger jumps occur when crossing the tool edge when measuring cutting forces.

In order to minimize the consumption of power energy related to the cutting forces achieved, it is recommended that the best combination of feed rate and depth of cut can be implemented.

## Figures and Tables

**Figure 1 materials-12-02864-f001:**
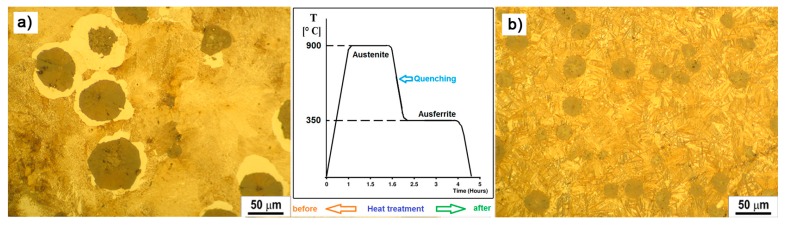
The microstructure of ductile iron (DI) castings EN-GJS-700-2 and austempered ductile iron (ADI) material (**a**) graphite nodules and microstructure of DI pearlite; (**b**) microstructure of ADI ausferrite.

**Figure 2 materials-12-02864-f002:**
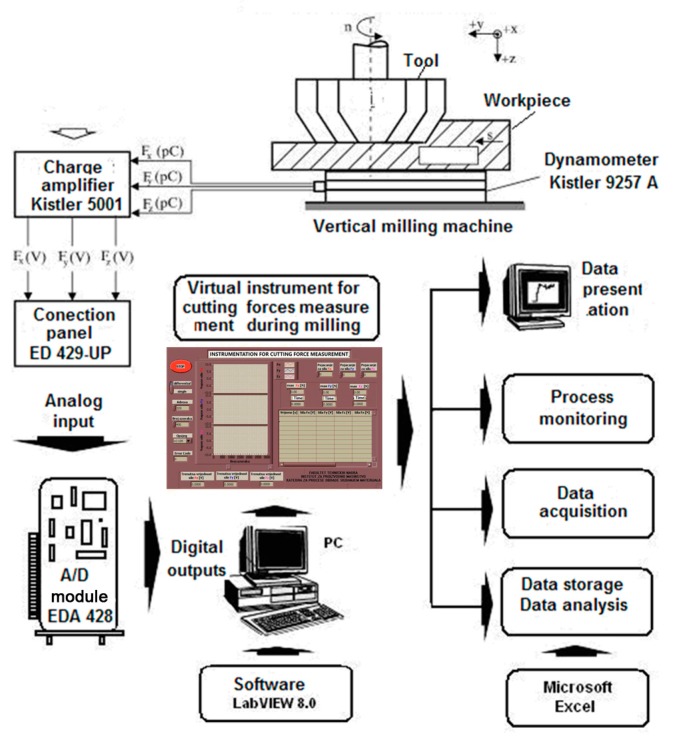
Model of acquisition system for monitoring of the three cutting force components.

**Figure 3 materials-12-02864-f003:**
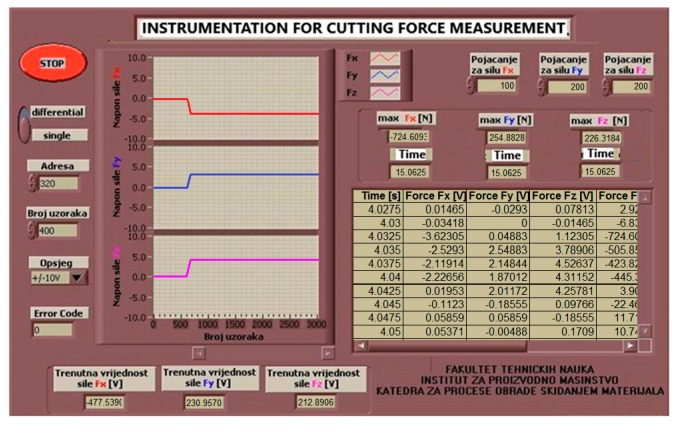
Front panel of virtual instrument.

**Figure 4 materials-12-02864-f004:**
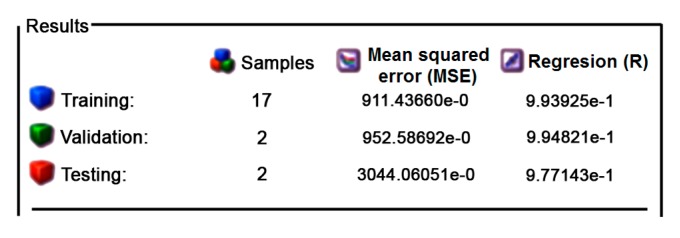
The relationship data for training, validation, and testing neural networks.

**Figure 5 materials-12-02864-f005:**
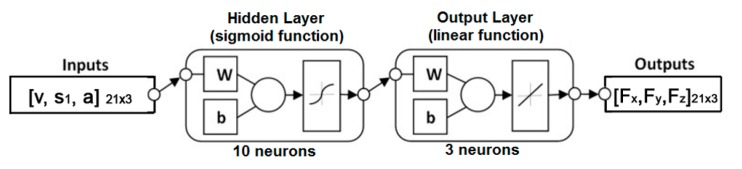
Architectural view of neural network flow chart.

**Figure 6 materials-12-02864-f006:**
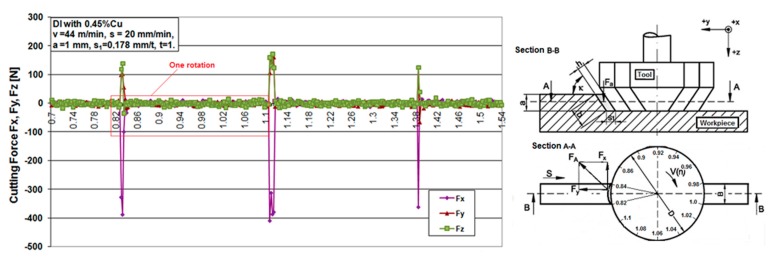
Cutting force components versus time for ductile iron.

**Figure 7 materials-12-02864-f007:**
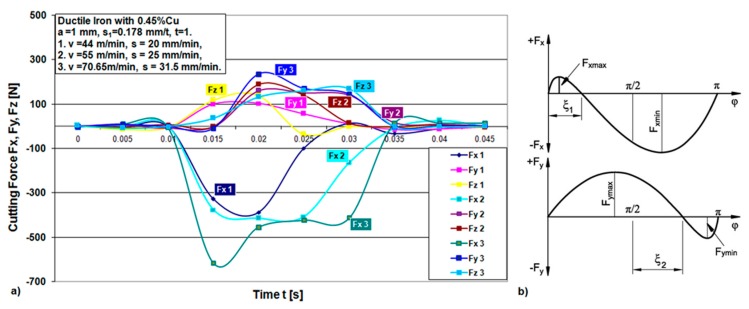
(**a**) Cutting force components time flow for one revolution during machining of ductile iron for three different speeds; (**b**) theoretical scheme of change of forces F_x_ and F_y_ [30].

**Figure 8 materials-12-02864-f008:**
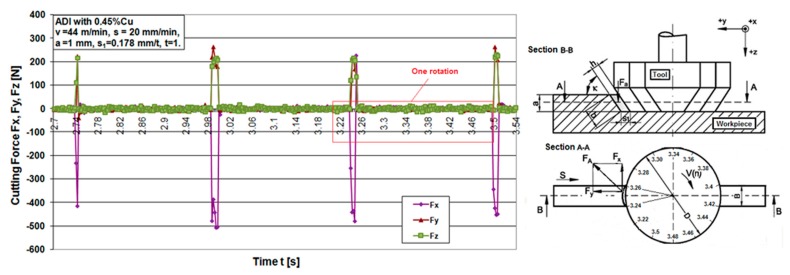
Cutting force components time flow during machining of ADI material.

**Figure 9 materials-12-02864-f009:**
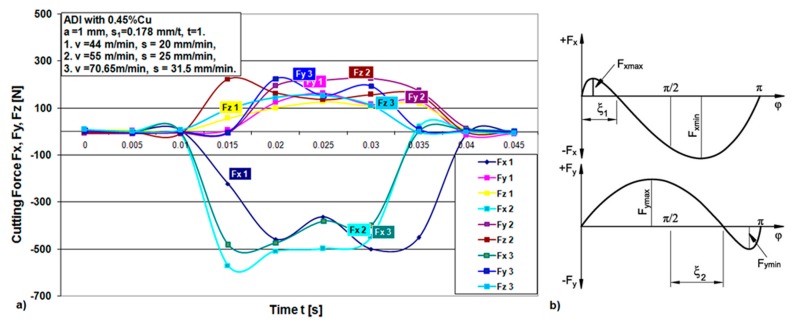
(**a**) Cutting force components time flow for one revolution during machining of ADI material for three different speeds; (**b**) theoretical scheme of change of forces F_x_ and F_y_ [30].

**Figure 10 materials-12-02864-f010:**
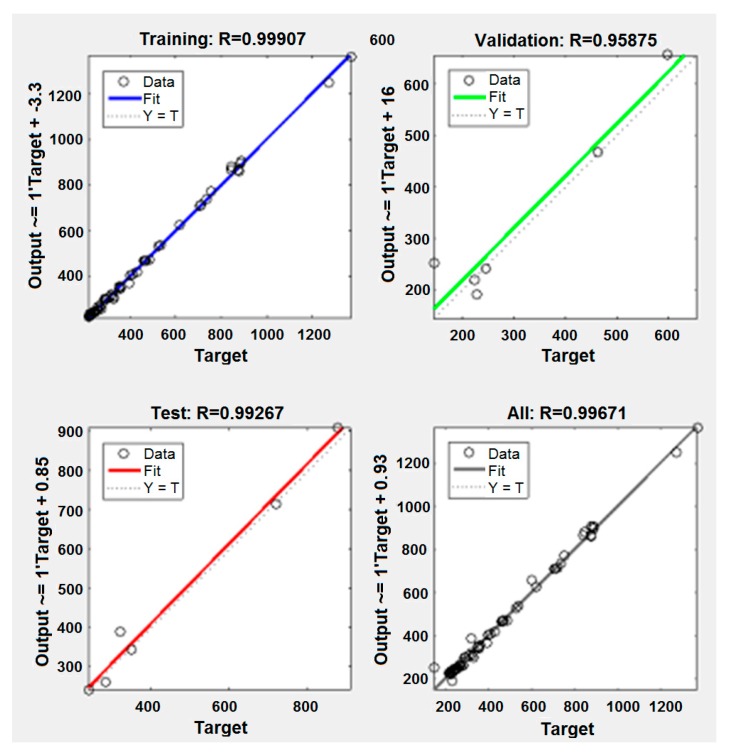
The regression coefficient (plot regression).

**Figure 11 materials-12-02864-f011:**
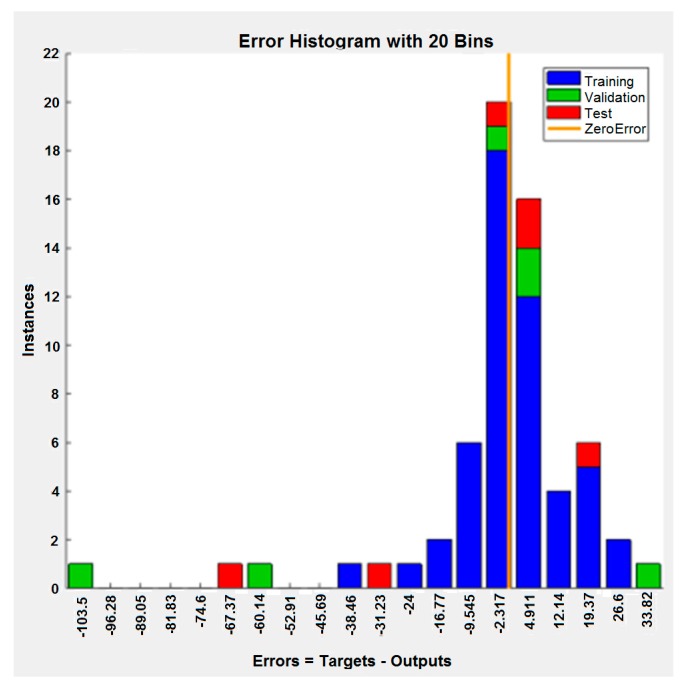
Histogram of errors of neural network created on the data from the simulation.

**Figure 12 materials-12-02864-f012:**
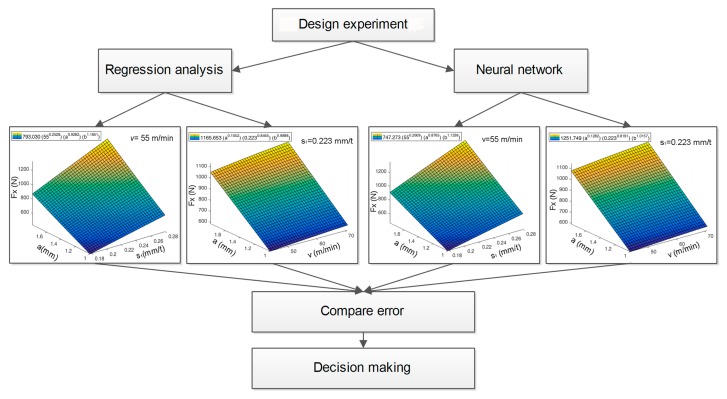
Comparing result of models.

**Figure 13 materials-12-02864-f013:**
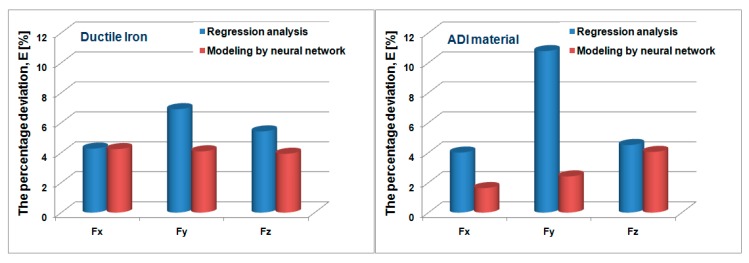
Percent deviation of the measured values in relation to the modeled values.

**Figure 14 materials-12-02864-f014:**
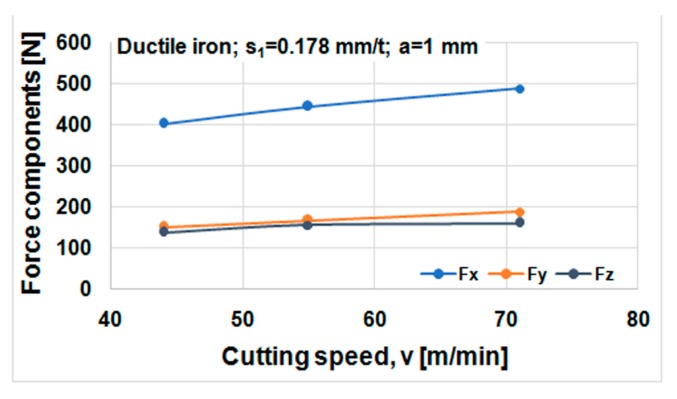
Cutting forces versus cutting speed, material DI.

**Figure 15 materials-12-02864-f015:**
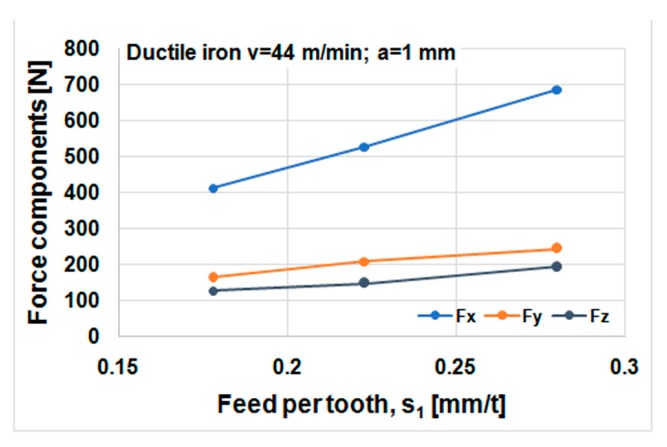
Cutting forces components against the feed per tooth, material DI.

**Figure 16 materials-12-02864-f016:**
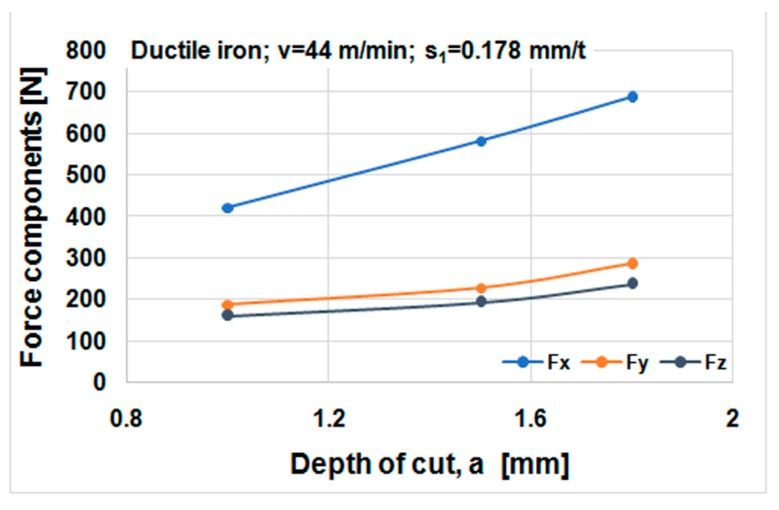
Cutting forces components against depth of cut, material DI.

**Figure 17 materials-12-02864-f017:**
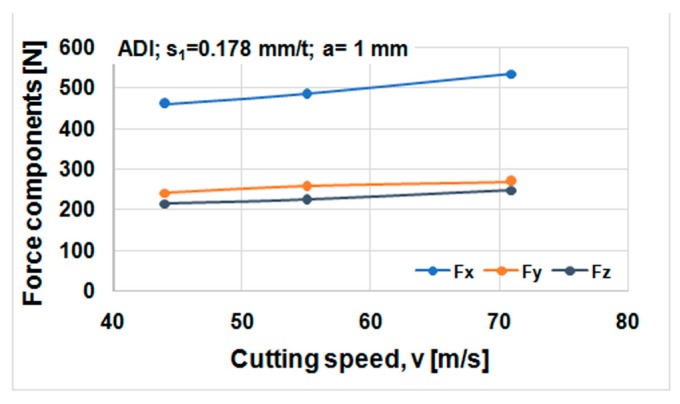
Cutting forces components against cutting speed, ADI material.

**Figure 18 materials-12-02864-f018:**
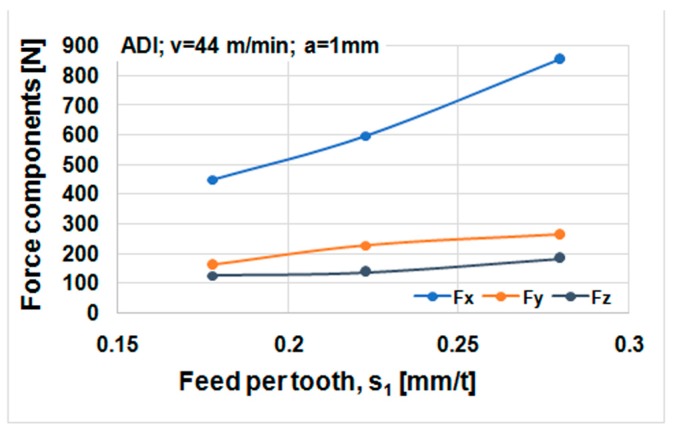
Cutting forces against feed per tooth, ADI material.

**Figure 19 materials-12-02864-f019:**
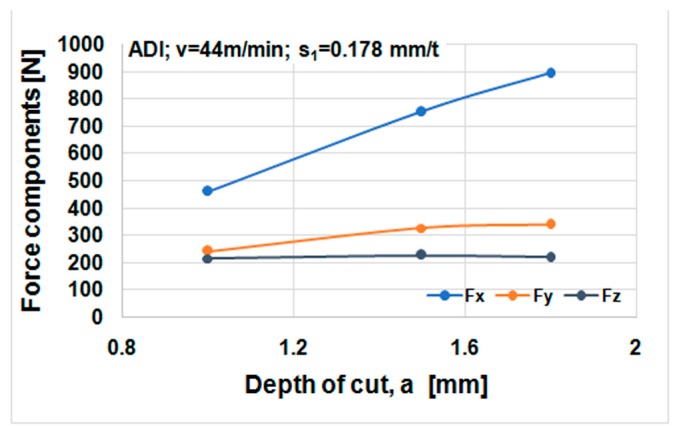
Cutting forces against depth of cut, ADI material.

**Table 1 materials-12-02864-t001:** Mechanical properties of materials in trial at room temperature.

Material	Tensile Strength R_m_ (MPa)	Yield Strength R_p0.2%_ (MPa)	Hardness HV_10_
DI	771	510	270
ADI	1110	995	480

**Table 2 materials-12-02864-t002:** Experimental cutting force components for different cutting regimes and researched materials.

No.	v (m/min)	s_1_ (mm/t)	a (mm)	DI			ADI		
				F_x_	F_y_	F_z_	F_x_	F_y_	F_z_
1	44	0.178	1	402	150	137	461	242	216
2	71	0.178	1	467	188	160	526	260	224
3	44	0.28	1	582	229	193	710	273	237
4	71	0.28	1	687	286	238	736	310	280
5	44	0.178	1.8	787	324	211	875	351	322
6	71	0.178	1.8	868	398	238	888	431	319
7	44	0.28	1.8	1246	466	345	1271	619	350
8	71	0.28	1.8	1349	565	416	1369	706	400
9	55	0.223	1.51	820	358	284	845	394	296
10	55	0.22	1.5	817	398	282	842	356	328
11	55	0.221	1.5	869	356	260	876	351	287
12	55	0.223	1.5	870	390	261	877	360	290
13	44	0.178	1.07	404	152	139	470	245	219
14	55	0.178	1	443	166	155	486	258	225
15	71	0.179	1	487	192	165	536	270	235
16	44	0.179	1	412	163	127	465	250	221
17	44	0.223	1	527	209	148	599	229	145
18	44	0.281	1	590	243	202	720	284	241
19	44	0.178	1.04	408	155	133	464	247	225
20	44	0.178	1.5	582	229	193	754	327	227
21	44	0.178	1.83	898	331	215	885	358	410

**Table 3 materials-12-02864-t003:** Constants in power cutting force models for DI.

Fi = C v^x^ s_1_^y^ a^z^				
Component	C	x	y	z
F_x_	793.030	0.2528	0.9282	1.1651
F_y_	137.600	0.4422	0.8590	1.2383
F_z_	202.117	0.3513	0.9878	0.8371

**Table 4 materials-12-02864-t004:** Constants in power cutting force models for ADI.

Fi = C v^x^ s_1_^y^ a^z^				
Component	C	x	y	z
F_x_	1165.6536	0.1552	0.8465	0.9898
F_y_	268.067	0.2799	0.7490	1.0714
F_z_	152.711	0.2925	0.4736	0.5515

**Table 5 materials-12-02864-t005:** Values of cutting force components determined by artificial neural network.

No.	DI			ADI		
	F_x_	F_y_	F_z_	F_x_	F_y_	F_z_
1	412.03	149.12	133.20	465.95	244.63	223.93
2	481.10	193.48	158.70	529.17	264.04	228.92
3	583.41	233.06	199.85	711.51	256.90	241.35
4	683.78	286.74	244.66	737.02	307.87	279.59
5	781.75	289.57	195.73	909.27	343.57	389.72
6	874.55	393.49	252.79	906.28	418.70	317.68
7	1243.10	470.69	347.03	1252.11	625.26	352.27
8	1337.90	567.16	404.46	1365.08	709.22	401.82
9	844.53	406.61	285.57	882.46	366.29	300.97
10	822.60	390.20	271.98	865.90	342.86	300.01
11	823.63	395.29	276.80	864.16	346.67	300.06
12	826.70	404.80	286.58	861.19	355.05	299.99
13	438.90	168.86	151.02	468.53	241.50	218.52
14	400.35	153.10	144.46	469.81	250.55	233.58
15	484.19	196.84	158.31	535.92	264.04	230.19
16	411.93	148.70	133.50	468.11	244.71	225.55
17	566.30	96.38	135.66	656.17	191.56	252.12
18	581.48	235.21	203.14	714.39	262.07	240.92
19	426.47	159.69	142.66	467.20	242.81	220.79
20	721.42	303.71	209.77	772.60	309.67	230.46
21	791.16	286.80	193.24	895.44	342.87	405.76

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
