# Peer review of "Comparative Characteristics of Ductile Iron and Austempered Ductile Iron Modeled by Neural Network"

_materials, 2019, doi:10.3390/ma12182864_

Round 1
Reviewer 1 Report
The paper topic is interesting and paper can be published but only after the above suggestions:
Abstract: The abstract is expected to include a brief digest of the research, that is, new methods, results, concepts, and conclusions only. The abstract needs to be more focused and achievements needs mentioned clearly. At the moment abstract is more like an introduction than abstract. Please add some information from the conclusion (quantifications).
Introduction generally based on old references. I personally feel that this part of paper is not concise enough from a reader’s perspective. Introduction must provide a comprehensive critical review of recent developments in a specific area or theme that is within the scope of the journal (advanced manufacturing), not only a list of published studies or a bibliometric one. Introduction is expected to have an extensive literature review followed by an in-depth and critical analysis of the state of the art. References section should be extensive about information connecting with machining mechanics. I suggest add information to better describe what other researchers have done in this area. I suggest add important and new articles from this field:
Intelligent optimization of hard turning parameters using evolutionary algorithms for smart manufacturing, Materials, 12, 879 (2019).
For what is fig 1?.
The discussion is shallow and needs more details, the observations and future trends. This chapter should be connected with others published papers.
Some of the bullet points on the conclusion are simplistic; Please try to emphasize your novelty, put some quantifications, and comment on the limitations. This is a very common way to write conclusions for a learned academic journal. The conclusions should highlight the novelty and advance in understanding presented in the work.
Additionally, Authors should present their work properly within the existing machining literature; i.e., papers published in the manufacturing and measurement journals, and compare their work with these latest related work. This comparison should be done analytically (in the introduction or related work section), experimentally (in the performance evaluation section), or both. Without this, this paper is not proper.
Author Response
Review 1
Comments and Suggestions for Authors
The paper topic is interesting and paper can be published but only after the above suggestions:
Abstract: The abstract is expected to include a brief digest of the research, that is, new methods, results, concepts, and conclusions only. The abstract needs to be more focused and achievements needs mentioned clearly. At the moment abstract is more like an introduction than abstract. Please add some information from the conclusion (quantifications).
Response 1: Please provide your response for Point 1. (in red)
The abstract was adjusted to the requirements of the reviewer.
Introduction generally based on old references. I personally feel that this part of paper is not concise enough from a reader’s perspective. Introduction must provide a comprehensive critical review of recent developments in a specific area or theme that is within the scope of the journal (advanced manufacturing), not only a list of published studies or a bibliometric one. Introduction is expected to have an extensive literature review followed by an in-depth and critical analysis of the state of the art. References section should be extensive about information connecting with machining mechanics. I suggest add information to better describe what other researchers have done in this area. I suggest add important and new articles from this field:
Intelligent optimization of hard turning parameters using evolutionary algorithms for smart manufacturing, Materials, 12, 879 (2019).
Response 2: Please provide your response for Point 2. (in red)
A critical review of recent developments in the field of research has been carried out, with new literature sources. An analysis is given with a major contribution to the areas of research.
For what is fig 1?.
Response 3: Please provide your response for Point 3. (in red)
The picture with the heat treatment process of obtaining ADI material from ductile cast iron was supplemented, which shows the difference in the microstructure of the tested materials.
The discussion is shallow and needs more details, the observations and future trends. This chapter should be connected with others published papers.
Response 4: Please provide your response for Point 4. (in red)
The discussion is broad and related to other published papers.
Some of the bullet points on the conclusion are simplistic; Please try to emphasize your novelty, put some quantifications, and comment on the limitations. This is a very common way to write conclusions for a learned academic journal. The conclusions should highlight the novelty and advance in understanding presented in the work.
Response 5: Please provide your response for Point 5. (in red)
The conclusion was expanded with the highlighted papers in the work, with comments on the limitations examined.
Additionally, Authors should present their work properly within the existing machining literature; i.e., papers published in the manufacturing and measurement journals, and compare their work with these latest related work. This comparison should be done analytically (in the introduction or related work section), experimentally (in the performance evaluation section), or both. Without this, this paper is not proper.
Response 6: Please provide your response for Point 6. (in red)
The paper is presented within the existing literature on the machining of ADI materials and the application of artificial neural networks; Comparison with the most recent papers was also made through the introduction, the experimental part and the discussion.

Reviewer 2 Report
The authors have used soft computing methods in order to investigate the influence of cutting parameters on cutting forces, when machining two different materials. The paper includes experimental data, however, several it lacks novelty and scientific rigor.
First of all, the authors need to rearrange the introduction section so that the references cited should explain the evolution of the research in this field and furthermore explain where the authors based their own work. Afterwards, a paragraph must follow that, based on the preceding analysis, explains what is the novelty presented in this paper and the possible applications (especially in connection to the used materials).
There are several Figures included in this paper that add little to the main subject of the paper; the authors need to reconsider their figures and present new, more meaningful ones, e.g 1, 3, 4 and 5-9 are providing very little information and are hard to read. The authors need to elaborate more on what is expected to understand from these figures, given that maximum forces can also be found in Tables.
The convergence of the NN models is quite good, however, as can be seen from Figures 14-19 the correlations are mostly linear. Thus the authors need to explain if their data is enough, the spectrum of the conditions is wide enough and if a method like NN is actually needed for such a correlation. It would also be useful to cross check the findings with similar results from the literature and make comparisons. The procedure of producing the NN also needs to be further explained as critical details are missing from the analysis.
The graphs in Figure 13 are hard to be compared; they need either to be combined in one graph or at least have the same maximum values in the axes. The discussion on the results is rather poor and trivial.
Finally, the conclusions need to be re-written and reflect the main findings of the research.
Author Response
Review 2
Comments and Suggestions for Authors
The authors have used soft computing methods in order to investigate the influence of cutting parameters on cutting forces, when machining two different materials. The paper includes experimental data, however, several it lacks novelty and scientific rigor.
First of all, the authors need to rearrange the introduction section so that the references cited should explain the evolution of the research in this field and furthermore explain where the authors based their own work. Afterwards, a paragraph must follow that, based on the preceding analysis, explains what is the novelty presented in this paper and the possible applications (especially in connection to the used materials).
Response 1: Please provide your response for Point 1. (in red)
The introductory section has been redrafted so that the cited references explain the development of research in this area on which the authors based their work. Subsequently, an analysis is given with the main contribution of the research.
There are several Figures included in this paper that add little to the main subject of the paper; the authors need to reconsider their figures and present new, more meaningful ones, e.g 1, 3, 4 and 5-9 are providing very little information and are hard to read. The authors need to elaborate more on what is expected to understand from these figures, given that maximum forces can also be found in Tables.
Response 2: Please provide your response for Point 2. (in red)
Figures 1, 3, 4 and 5-9 have been corrected and supplemented with information to make it more readable to the reader. The authors have detailed in the text what is displayed with these Figures.
The convergence of the NN models is quite good, however, as can be seen from Figures 14-19 the correlations are mostly linear. Thus the authors need to explain if their data is enough, the spectrum of the conditions is wide enough and if a method like NN is actually needed for such a correlation. It would also be useful to cross check the findings with similar results from the literature and make comparisons. The procedure of producing the NN also needs to be further explained as critical details are missing from the analysis.
Response 3: Please provide your response for Point 3. (in red)
A comparison of the regression results obtained by ANN with other literature sources was performed to demonstrate the validity of their application. An explanation of the contribution of ANN in terms of data processing and subsequent optimization is given, which is not entirely possible with the diagrams presented in Figures 14-19, which show the one-factor dependencies of the examined parameters. An ANN extension is possible with three influential factors to four if the need arises.
The graphs in Figure 13 are hard to be compared; they need either to be combined in one graph or at least have the same maximum values in the axes. The discussion on the results is rather poor and trivial.
Response 4: Please provide your response for Point 4. (in red)
The graphs in Figure 13 have been corrected so that they have the same maximum values in the axes and the analysis in the graphs is performed. There has been widespread discussion about the results of the research.
Finally, the conclusions need to be re-written and reflect the main findings of the research.
Response 5: Please provide your response for Point 5. (in red)
The conclusion was expanded with the highlighted novelty in the paper.

Round 2
Reviewer 1 Report
Paper is ready for publication, congratulations for Authors.
Author Response
Dear Editor of the Materials,
Please find enclosed our revised paper with the list of changes.
We have corrected all the remarks:
The title should not contain any abbreviation: ADI should be spelled out.
Response: An abbreviation in the title is excluded.
Same for the Abstract, in which there is more: ANN.
Response: In the abstract all ANN are replaced.
The English of the paper is NOT acceptable. Almost each sentence needs proper styling. There is no problem with the scientific content.
Response: Spell check English and style was done.
The authors would like to thank you for your suggestions which helped us greatly in our effort to revise the paper.

Reviewer 2 Report
The authors have considerably improved their manuscript according to the reviewer's comments.
The paper can be published as is.
Author Response

(The authors gave the same response as above.)
